# Taqman qPCR Quantification and *Fusarium* Community Analysis to Evaluate Toxigenic Fungi in Cereals

**DOI:** 10.3390/toxins14010045

**Published:** 2022-01-06

**Authors:** Elina Sohlberg, Vertti Virkajärvi, Päivi Parikka, Sari Rämö, Arja Laitila, Tuija Sarlin

**Affiliations:** 1VTT Technical Research Centre of Finland Ltd., FI-02044 Espoo, Finland; vertti@no-office.co (V.V.); Arja.Laitila@outlook.com (A.L.); tuija.sarlin@vtt.fi (T.S.); 2Natural Resources Institute Finland (Luke), FI-31600 Jokioinen, Finland; ext.paivi.parikka@luke.fi (P.P.); sari.ramo@luke.fi (S.R.)

**Keywords:** *Fusarium*, mycotoxins, DON, qPCR, NGS, Fusarium head blight, cereals

## Abstract

Fusarium head blight (FHB) is an economically important plant disease. Some *Fusarium* species produce mycotoxins that cause food safety concerns for both humans and animals. One especially important mycotoxin-producing fungus causing FHB is *Fusarium graminearum*. However, *Fusarium* species form a disease complex where different *Fusarium* species co-occur in the infected cereals. Effective management strategies for FHB are needed. Development of the management tools requires information about the diversity and abundance of the whole *Fusarium* community. Molecular quantification assays for detecting individual *Fusarium* species and subgroups exist, but a method for the detection and quantification of the whole *Fusarium* group is still lacking. In this study, a new TaqMan-based qPCR method (FusE) targeting the *Fusarium*-specific elongation factor region (*EF1α*) was developed for the detection and quantification of *Fusarium* spp. The FusE method was proven as a sensitive method with a detection limit of 1 pg of *Fusarium* DNA. *Fusarium* abundance results from oat samples correlated significantly with deoxynivalenol (DON) toxin content. In addition, the whole *Fusarium* community in Finnish oat samples was characterized with a new metabarcoding method. A shift from *F. culmorum* to *F. graminearum* in FHB-infected oats has been detected in Europe, and the results of this study confirm that. These new molecular methods can be applied in the assessment of the *Fusarium* community and mycotoxin risk in cereals. Knowledge gained from the *Fusarium* community analyses can be applied in developing and selecting effective management strategies for FHB.

## 1. Introduction

Fusarium head blight (FHB), a devastating plant disease that infects wheat, barley, oats, and other small grain cereals, causes billions of dollars of crop yield and quality losses worldwide [1,2,3]. Up to 17 fungal species have been shown to cause FHB, among which most species belong to the genus *Fusarium*, and a couple of species belong to genus *Microdochium* [4,5]. Often, *Fusarium* species form a disease complex where different *Fusarium* species co-occur in the infected cereals [6]. Some *Fusarium* species produce mycotoxins that cause a risk for the end-use of the cereal crops in the food and feed industry. One especially important plant pathogen is *Fusarium graminearum*, which produces the mycotoxins trichothecenes deoxynivalenol (DON) and nivalenol (NIV) and the mycoestrogen zearalenone (ZEN) [7]. The identification of species that inflict FHB is important, since they differ in fungicide sensitivity, host resistance, pathogenicity, and toxin production [8]. In Finnish cereals, the most common *Fusarium* species detected in 2005–2014 were *F. avenaceum*, *F. culmorum*, *F. graminearum*, *F. poae*, *F. sporotrichioides*, and *F. langsethiae* [9]. The prevalence of *F. culmorum* and *F. graminearum* was high in oats and barley, but when compared to previous studies, a shift from *F. culmorum* to *F. graminearum* was detected, which has led to increased DON content. A rapid increase in the prevalence of *F. graminearum* has also been detected in other European countries [10,11,12]. Effective FHB management strategies are needed and the development of these management methods requires information about the composition and abundance of the whole *Fusarium* community, because different populations could be affected selectively [13]. To evaluate the impact of fungal control means in disease control, quantitative data of the *Fusarium* community and different species are needed.

A variety of different methods are currently applied in the detection and identification of *Fusarium* species in cereals—for a recent review, see Karelov et al. [14]. Traditional plating methods on a selective nutrient medium applied in the detection of *Fusarium* fungi are laborious and some species can outgrow others, preventing the detection of the whole *Fusarium* community. In the past few years, the focus in *Fusarium* detection has been on molecular DNA-based methods. The most prevalent DNA-based methods for *Fusarium* quantification and identification are single and multiplex qPCR methods that detect different *Fusarium*-specific DNA regions. Bluhm et al. [15] developed a TaqMan real-time PCR assay for group-specific detection of trichothecene- and fumonisin-producing *Fusarium* spp. by targeting the genes involved in mycotoxin biosynthesis, and Sarlin et al. [16] evaluated the application of the method of *Fusarium* trichothecene gene-region measurement to detect toxigenic fungi in barley and malt. In addition, Kulik et al. [17] developed a TaqMan-based qPCR assay targeting the enniatin synthetase gene to quantify *Fusarium avenaceum*, *Fusarium tricinctum*, and *Fusarium poae* abundance in wheat grains, and a positive and significant correlation was observed between the *Fusarium* amount and the total concentration of enniatins. Waalwijk et al. [8] reported the development and application of a TaqMan qPCR method to quantify individual *Fusarium* species and closely related *Microdochium nivale* in infected cereals. Recently, Morcia et al. [5] applied a new chip digital PCR method for quantification of *F. graminearum*, *F. culmorum*, *F. sporotrichioides*, *F. poae*, and *F. avenaceum*. Then, Sonia et al. [17] developed individual qPCR methods for seven *Fusarium* and two *Microdochium* species. The co-occurrence of several *Fusarium* species in cereals makes monitoring more challenging, and the detection of only a few *Fusarium* species is not sufficient. Genus-specific conventional PCR methods are available for the detection of the presence of *Fusarium* fungi in cereals [14], but a genus-specific quantitative PCR method for quantification of all the *Fusarium* species is still lacking.

New metabarcoding methods using the *Fusarium* elongation factor region (*EF1α* gene) have been applied in assessing the diversity of the *Fusarium* community in cereals [18,19]. These new metabarcoding methods enable the detection of the *Fusarium* community to the species level and monitoring the distribution and community shifts in cereals in changing environmental conditions. In addition, new emerging species can be detected with these methods.

In this study, a new TaqMan-based qPCR method (FusE) targeting the *Fusarium*-specific elongation factor region (*EF1α*) was developed for the detection and quantification of whole *Fusarium* spp. simultaneously. As a proof of principle, new FusE qPCR analyses were performed on field samples to assess the *Fusarium* contamination in Finnish oat crop survey samples collected in 2015–2017. The years 2016 and 2017 were especially challenging regarding the *Fusarium* contamination, and 25% of the national yearly crop survey oat samples collected by the Finnish Cereal Committee had mycotoxin deoxynivalenol (DON) content over 1750 µg/kg. To evaluate the connection between *Fusarium* abundance and DON content, the correlation between the FusE qPCR, *Fusarium* plating method, and deoxynivalenol (DON) results was evaluated. In addition, the whole *Fusarium* community in oat samples was characterized with a novel *Fusarium* metabarcoding method.

## 2. Results

### 2.1. FusE qPCR Method

The amplification with the newly designed FusE primers and probe confirmed a highly specific amplification, and only one single product with a size of 96 bp was amplified with the FusE primers. All 14 *Fusarium* species tested (Table 1) were amplified with the primer set. Other tested fungi, yeast, and bacteria associated and isolated from cereal samples (Table 1) were not detected with the primes. In addition to *Fusarium* species, *Microdochium nivale*, and *M. majus* strains were amplified with FusE primers.

Linearity, efficiency, and the theoretical limit of detection were estimated from the standard curve with *Fusarium* genomic DNA ranging from 1 to 0.001 ng (Figure 1). The cycle threshold (Ct) values from the standard curve showed a linear dynamic range, corresponding to a Ct range of 23 to 34 cycles. The lower detection limit of the *Fusarium* spp. was 0.001 ng of *Fusarium* genomic DNA per reaction, as 35 cycles was set to be the cutoff value due to uncertainty of the last five cycles. Linear regression between the amount of *Fusarium* DNA (ng) and the corresponding Ct values revealed r^2^ values of 1, while the mean efficiency of the qPCR was 95% (*n* = 10). No PCR inhibition was observed when a known amount of *Fusarium* genomic DNA was added to the DNA sample containing the cereal matrix (data not shown).

### 2.2. Deoxynivalenol Levels in Oat Samples

The deoxynivalenol (DON) content in oat grain varied substantially between the samples collected in different years (Figure 2). In 2015, the DON content ranged from 0 to 2700 µg/kg, and the average (*n* = 9) was 1103 µg/kg. In 2016 and 2017, the higher DON contamination of grain was detected, and the content of this mycotoxin varied significantly between the samples. In 2016, the DON content ranged from 130 to 26,000 µg/kg and the average (*n* = 27) was 5334 µg/kg, whereas in 2017, the DON content varied between 310 and 23,000 µg/kg and the average (*n* = 15) was 6620 127 µg/kg.

### 2.3. Fusarium Abundance in Field Samples by a Conventional Plating Method

Various *Fusarium* species were detected from oat samples in the years 2015–2017 by the plating method (Figure 3). The most abundant fungi were *F. graminearum* and *F. avenaceum*. In the year 2015, the percentage of *F. graminearum*- and *F. avenaceum*-contaminated grains in oat samples varied from 20% to 56%, with an average (*n* = 9) of 35%, and from 2% to 31%, with an average of 12%, respectively. Other minor *Fusarium* species detected in 2015 were *F. arthrosporioides*, *F. culmorum*, *F. tricinctum*, *F. poae*, *F. sporotrichioides*, *F. langsethiae*, and *F. sambucinum*. In 2016, the *F. graminearum* contamination rate varied between 2% and 87%, with an average (*n* = 27) of 51%, and the *F. avenaceum* contamination rate varied between 3% and 55%, with an average of 25%. Other minor *Fusarium* species cultivated were *F. arthrosporioides*, *F. culmorum*, *F. tricinctum*, *F. poae*, *F. sporotrichioides*, *F. langsethiae*, *F. equiseti*, and *F. sambucinum*. In 2017, the percentage of grains contaminated with *F. gramiearum* plating method varied between 27% and 80%, with an average (*n* = 15) of 61%, and the percentage of those contaminated with *F. aveneceum* varied between 0% and 28%, with an average of 8%. Other minor *Fusarium* species detected by the plating method were *F. arthrosporioides*, *F. culmorum*, *F. tricinctum*, *F. poae*, *F. sporotrichioides*, *F. langsethiae*, and *F. sambucinum*.

### 2.4. Fusarium Abundance Detected with FusE qPCR

Low *Fusarium* DNA levels in field samples ranging from 0.3 to 2.4 pg of *Fusarium* DNA ng^−1^ of total DNA with average (*n* = 9) of 1.1 ± 0.6 pg of *Fusarium* DNA ng^−1^ of total DNA were detected in 2015 (Figure 4). *F. graminearum* and *F. culmorum* levels were not measured from the 2015 field samples, because of low *Fusarium* spp. contamination levels. In 2016 and 2017, cereal cultivation conditions promoted *Fusarium* growth, and increased *Fusarium* abundance was measured. The *Fusarium* DNA levels in oat samples in 2016 and 2017 were triple compared to the year 2015. In 2016, the *Fusarium* spp. DNA levels ranged from 1.3 to 16.1 pg of *Fusarium* DNA ng^−1^ of total DNA with an average (*n* = 27) of 5.3 ± 3.7 pg of *Fusarium* DNA ng^−1^ of total DNA. In 2017, the *Fusarium* spp. abundance ranged from 3.2 to 15.5, with an average (*n* = 15) of 8.5 ± 3.9 pg of *Fusarium* DNA ng^−1^ of total DNA. The variation of the *Fusarium* DNA levels was substantial between the samples in 2016 and 2017. *F. graminearum* abundance in samples from the year 2016 was high, ranging from 0.4 to 16.3, with an average (*n* = 27) of 3.6 ± 3.5 pg of *F. graminearum* DNA ng^−1^ of total DNA. In addition, small amounts of *F. culmorum* were detected in some of the samples, ranging from 0.0001 to 0.24 pg of *F. culmorum* DNA ng^−1^ of total DNA. In the year 2017, high amounts of *F. graminearum* DNA were detected again, ranging from 3.2 to 8.1, with an average (*n* = 15) of 2.6 ± 2.2 pg of *F. graminearum* DNA ng^−1^ of total DNA. No *F. culmorum* was detected in the samples collected in 2017.

### 2.5. Fusarium Community in Field Samples

The *Fusarium* community diversity in oat samples was analyzed with next-generation sequencing. A total number of 893,699 sequences of the *Fusarium* elongation factor region were obtained from Illumina Miseq (Illumina, Inc., San Diego, CA, USA) sequencing after quality control. These sequences were divided into 420 amplicon sequence variants (ASVs), from which 267 ASVs belonged to the genus *Fusarium*. The dominant species was *Microdochium nivale*, which is a close relative to *Fusarium* species and has previously been associated with the *Fusarium* genus (Appendix A). Species that did not belong to the *Fusarium* genus were extracted from the data to study the *Fusarium* community in more detail. The *Fusarium* community structure varied between the years (Figure 5). Based on the community profiling, the most abundant species in the 2015 oat crop was *F. oxysporum*, with relative abundance ranging from 9% to 99% of the total community in the samples. In one sample (VV16_38), the dominant species was *F. avenaceum* (76%). In 2015, *F. graminearum* was found as a minor group, and relative abundance ranged from 0 to 26%. Other *Fusarium* species detected in 2015 were *F. poae* and *F. langsethiae*. In 2016, the dominant *Fusarium* species were *F. oxysporum* (2–87%) and *F. graminearum* (0–87%). Other *Fusarium* species detected were *F. avenaceum*, *F. langsethiae*, *F. poae*, *F. tricinctum*, and *F. equiseti*. In addition, strain NRRL 25,130 detected in sample VV17-13 has been identified as *Fusarium avenaceum* [20]. In 2017, *F. oxysporum* was the dominant *Fusarium* species, and the relative abundance ranged from 1.5% to 82%, but *F. graminearum* was also detected in the most of samples (0–88%), and was the dominant species in two oat samples. Other *Fusarium* species detected from the oat samples were *F. poae*, *F. avenaceum*, and *F. langsethiae*.

### 2.6. Correlation of Fusarium Abundance and DON Levels

The correlation of the *Fusarium* abundance in oat samples detected with the FusE qPCR method and DON content was analyzed. In addition, the correlation of qPCR results and *Fusarium* abundance detected with the plating method was analyzed. The correlation was calculated from the whole sample set consisting of 157 oat samples and 40 barley samples from which the FusE assay, species-specific qPCR assays, a conventional *Fusarium* plating method, and DON measurement were performed. Based on the statistical analysis, the DON content correlated significantly (*p*-value < 0.01) with the amount of *Fusarium* spp. DNA detected with the FusE qPCR assay and with the amount of *F. graminearum* DNA detected with the *F. graminearum* specific qPCR assay in all three years studied (Figure 6). *Fusarium* spp. qPCR results and *F. graminearum* qPCR results also correlated significantly (*p*-value < 0.01) with the *F. graminearum* plating method results.

## 3. Discussion

An efficient and rapid method for *Fusarium* quantification is needed for assessing the FHB and mycotoxin risk in cereals. Quantitative PCR assays with TaqMan-based primers for specific individual *Fusarium* species and some toxin-producing *Fusarium* species have been developed [8,15,17]. However, an efficient method for quantifying the whole *Fusarium* group with one qPCR method has been lacking so far. The co-occurrence of several *Fusarium* species in the FHB complex in cereals makes monitoring challenging. If only a few *Fusarium* species are targeted with qPCR, the *Fusarium* contamination can remain underestimated or even undetected.

This study developed a new TaqMan qPCR assay targeting the *Fusarium* elongation factor region (*EF1α*) to quantify the abundance of *Fusarium* spp. in cereal samples. To validate the method with representative material, DNA from 157 oat samples and 40 barley samples were extracted. The DNA extraction method applied for the oat samples was successfully used for cereal samples including barley in a previous study by Sarlin et al. [16]. Samples were analyzed with the FusE qPCR assay, *F. graminearium*, and *F. culmorum*-specific qPCR assays, conventional *Fusarium* plating method, and DON measurement. The correlation of the FusE method with DON content was analyzed from the whole sample set. The *Fusarium* metabarcoding method was applied to a subset of 51 oat samples to which both high and low *Fusarium*-contaminated samples were selected from each year. Although we applied the FusE qPCR assay for cereal samples in this study, the method can be used to quantify *Fusarium* DNA in many different environmental samples (data not shown). The specificity of our assay was confirmed in silico, and subsequently against a diverse panel of fungal and bacterial species associated with cereals. Only *Fusarium* species, as well as *Microdochium nivale* and *Microdochium majus*, were amplified with the FusE qPCR assay. The cross-reaction between *M. nivale* strains was expected, since *M. nivale* is a close relative to *Fusarium* fungi and has previously been part of the *Fusarium* genus [21,22]. Cross-reaction was not detected with DNA extracted from sterile oat cells, revealing that the primers were not amplifying any regions of oat DNA. Furthermore, PCR-inhibitory effects were not detected when spiking a DNA sample isolated from oat grains with a known amount of *Fusarium* DNA.

The FusE qPCR method developed in this study proved as an effective tool to monitor *Fusarium* contamination in cereals. The FusE qPCR assay revealed that the amount of *Fusarium* DNA and thus the severity of *Fusarium* infection was notably higher in cereal samples collected in the years 2016 and 2017 compared to the samples collected in 2015. The amounts of *Fusarium* spp. DNA and *F. graminearum* DNA analyzed with the FusE and *F. graminearum* species-specific qPCR assays, respectively, correlated significantly with deoxynivalenol (DON) toxin content in each sampling year. *F. culmorum* DNA levels were very low in all years and no correlation was found between *F. culmorum* qPCR results and DON content. FusE qPCR method can be used to quantify the total *Fusarium* biomass to screen the mycotoxin risk in cereal samples. If *Fusarium* DNA is detected, then the presence and abundance of specific toxin-producing *Fusarium* species can be measured using species-specific *Fusarium* primers, such as *F. graminearum* or *F. culmorum* primers, to evaluate the mycotoxin risk more precisely [8,23]. In our study, the high correlation between the amounts of *Fusarium* DNA and DON content was detected, but in a study by Sarlin et al. [16], barley grain samples contained relatively high amounts of trichothecene-producing *Fusarium* DNA, although no DON was detected. Some of the results presented by Sarlin et al. [16] were explained with the production of other trichothecenes than DON and low levels of *F. graminearum*, but also environmental factors such as temperature, humidity, and other microbes, and the host plant can influence the mycotoxin production of *Fusarium* species in cereals [24,25]. The effects of environmental factors can impact the estimation of mycotoxin risk based only on the *Fusarium* DNA levels. Another application for the FusE method is the assessment of the gushing risk in malting barley such as that studied by Virkajärvi et al. [26], where a higher *Fusarium* DNA level was associated with a higher gushing risk.

The economic losses caused by FHB to cereal producers are significant, and thus effective management strategies for FHB are needed. Important tools to monitor the FHB are *Fusarium*-specific detection and quantification methods. With these methods, the effectiveness of the management methods can be evaluated and mycotoxin risk in cereals can be assessed. In addition to the level of *Fusarium* contamination in cereals, the development of these management methods requires information about the composition of the whole *Fusarium* community because different populations could be affected selectively [13]. In this study, we evaluated the suitability of the new *Fusarium* metabarcoding method by Cobo-Díaz et al. [19] for *Fusarium* community profiling in cereal samples. With this new metabarcoding method, the whole *Fusarium* community in Finnish oat samples was characterized. Similar *Fusarium* species were detected with both the *Fusarium* plating method and metabarcoding methods. *F. graminearum* was detected especially in the oat samples collected in 2016 and 2017 according to both the plating method and metabarcoding methods. Only low amounts of *F. culmorum* were detected in the oat samples with plating and metabarcoding methods. *F. oxysprorum* was detected as the dominant *Fusarium* species in Finnish oat samples with metabarcoding. *F. oxysporum* was also the most abundant species in soil samples collected from maize fields in the study of Cobo et al. where the *Fusarium* metabarcoding method was established [19]. *F. oxysporum* was not detected in oat samples with the *Fusarium* plating method, and *F. arthrosporioides* and *F. sambucinum* were detected from oat samples by the plating method but not by the metabarcoding method. The plating method can favor some *Fusarium* species over others, and species growing faster can overtake other slower growing species, which can explain the differences in the results by the plating method and metabarcoding method. In addition, some sequences could not be identified to the species level with the metabarcoding method. This could be due to the lack of *Fusarium* elongator factor sequences in the sequence databases, or the fact that these species cannot be distinguished from each other because of sequence similarity in the elongation factor region between those taxa [19].

A shift from *F. culmorum* to *F. graminearum* in FHB infected cereal grains has been detected in Europe [10,11,12]. The results of this study confirm that *F. graminearum* was detected as the main mycotoxin-producing *Fusarium* species in Finnish oat samples. The reason for this shift is still partly unknown, but Parikka et al. [27] indicated that the shift is due to climate changes and farming practices. Climate is known to affect the *Fusarium* incidence and *Fusarium* community structure [28].

New *Fusarium* metabarcoding methods enable the detection of the *Fusarium* community to the species level and monitoring the distribution and community shifts in cereal production. In addition, new emerging species can be detected with these methods. Compared to the laborious and time-consuming plating method that needs high levels of expertise in *Fusarium* species identification, the new *Fusarium* metabarcoding method provides the *Fusarium* species information from cereal samples faster and more reliably. The *Fusarium* metabarcoding method by Cobo-Díaz et al. [19] detected *Fusarium* species as well as *Microdochium nivale*. *M. nivale* is a close relative to *Fusarium* species and has been previously part of the *Fusarium* genus, which explains the similarities in the targeted elongation factor region and amplification of *M. nivale* with primers Fa-150 and Ra2 designed by Cobo-Díaz et al. [19]. The *Fusarium* plating method is only a semi-quantitative method because the method only shows the percentage of the contaminated grains. With the *Fusarium* metabarcoding method, the relative abundance of the *Fusarium* species is obtained. Relative abundances that are currently used in metabarcoding studies can be compared within the same dataset, but this can lead to misinterpretations of microbial community structures. Because the sequencing data are compositional, the increase in one taxon leads to the concurrent decrease in the others [29]. Quantitative PCR provides the absolute quantification of the *Fusarium* DNA but does not give the species identification, and developing a species-specific qPCR method for all *Fusarium* species is not reasonable. Quantitative PCR combined with metabarcoding could be used in the absolute quantification of microbial abundances [29]. FusE qPCR assay combined with *Fusarium* metabarcoding method could be used in the absolute quantification of the whole *Fusarium* species complex in cereal samples.

*Fusarium* community structure in cereals can be studied in more detail using the novel *Fusarium* metabarcoding method. The significance of the whole cereal microbiome for the development of *Fusarium*-related diseases has been recognized lately. Both interactions within the *Fusarium* microbiome and interaction with the plant microbiome in the cereals play an important role in the FHB outbreaks and accumulation of mycotoxins [30]. Knowledge gained from these interactions can be used in developing new management methods for FHB. Future studies are especially needed to identify the characteristics of cereal microbiomes linked to *Fusarium* suppression.

## 4. Materials and Methods

### 4.1. Materials

The fungal and bacteria pure cultures used in this study were obtained from the culture collection of VTT Technical Research Centre of Finland Ltd. (VTT Culture Collection, Espoo, Finland) (Table 1). From the *Fusarium* genus, 14 different *Fusarium* species were used in the evaluation of the specificity of the developed FusE qPCR method. In addition, 12 other fungal strains commonly associated with cereals, one yeast strain, and two bacterial strains were tested. Fungal strains were cultured on potato dextrose agar (CM0139, Oxoid Ltd., Hampshire, UK), yeast strain on yeast mold agar (yeast: B271210, Becton, Dickinson and Company, Franklin Lakes, NJ, USA), and bacterial strains on MRS agar (de Man, Rogosa and Sharpe, CM0361, Oxoid Ltd., Hampshire, UK). Strains were incubated for 2–4 days at 25 °C before the DNA was isolated.

### 4.2. Field Samples

Cereal samples were selected among the yearly crop survey samples collected around Finland by the Finnish Cereal Committee during the years 2015–2017. The total amount of the samples consisted of 157 oat samples and 40 barley samples. DON content detected in the cereal samples ranged from <25 to 26,000 µg/kg.

In this paper, a subset of 51 oat samples was selected from the whole field sample set to compare the results of the new *Fusarium*-specific FusE qPCR method, the conventional *Fusarium* plating method, *F. graminearum*- and *F. culmorum*-specific qPCR methods and DON measurement. Both samples with high and low *Fusarium* contamination were selected in the subset. In addition, the new *Fusarium*-specific metabarcoding method was applied for the same subset of 51 oat samples. The correlation analysis was performed for the whole field sample set.

### 4.3. DNA Isolation

DNA from pure microbial isolates and ground cereal samples was extracted with FastDNA^®^ SPIN Kit for Soil (MP Biomedicals, Irvine, CA, USA). Cereal samples (10–20 g) were ground into fine flour with a grain mill (Bauknecht, Stuttgart, Germany). The DNA extraction was carried out from 100 mg of microbial biomass of the pure cultures or grain flours according to the manufacturer’s instructions but using a lysing step modified by VTT. The lysis was performed for 2 × 1 min at 6.0 s^−1^ using the FastPrep^®^ Cell Disrupter (MP Biomedicals, Irvine, CA, USA) and Lysing Matrix A tubes (MP Biomedicals, Irvine, CA, USA). The tubes were placed on ice in between the two lysis runs. Centrifugation steps in the DNA extraction were carried out in Eppendorf Centrifuge 5424 (Eppendorf, Hamburg, Germany) and 13,000× *g*. DNA was extracted in triplicate from each cereal sample. Extracted DNA was stored in a freezer at −20 °C until being further analyzed.

### 4.4. Design of Primers and Hydrolysis Probe for Fusarium Genus

Primers FusEF and FusER and probe FusEP for the FusE TaqMan qPCR were designed in silico (Table 2). The reference *Fusarium* elongation factor *EF1α* gene sequences were searched from the JGI database (Joint Genome Institute, https://genome.jgi.doe.gov/portal/, accessed on 1 June 2017), Broad Institute (https://www.broadinstitute.org/, accessed on 1 June 2017), and Ensemble Fungi (https://fungi.ensembl.org/index.html, accessed on 1 June 2017). Specific regions to *Fusarium* species were identified using BLASTn search (NCBI, Bethesda, MD, USA, https://blast.ncbi.nlm.nih.gov/Blast.cgi, accessed on 1 June 2017) against the NCBI nucleotide collection (nr/nt) (NCBI, Bethesda MD, USA, https://www.ncbi.nlm.nih.gov/, accessed on 1 June 2017). Next, ClustalO (The European Bioinformatics Institute (EMBL-EBI), Cambridgeshire, United Kingdom https://www.ebi.ac.uk/Tools/msa/clustalo/, accessed on 1 June 2017) was used to align ten closest homologs. Several primer pairs binding to the *Fusarium EF1α* gene were designed using Primer-BLAST (NCBI, Bethesda MD, USA, https://www.ncbi.nlm.nih.gov/tools/primer-blast/, accessed on 1 June 2017) and the NCBI nr database and a primer pair that amplified only *Fusarium* species were selected. The designed primers amplified a 96 bp region of the *EF1α* gene, and the melting temperature of the primers was 59–68 °C. The primers and the probe used in this study were synthesized by Integrated DNA Technologies (IDT Inc., Coralville, IA, USA). A reporter dye FAM (6-carboxy-fluorescein) was used in labeling the 5′ end and quencher dye TAMRA (6-carboxy-tetramethyl rhodamine) the 3′ end of the probe FusEP.

### 4.5. TaqMan qPCR and Data Analysis

qPCR amplification was performed in the LightCycler^®^ 480 qPCR device (Roche Diagnostics Ltd., Risch-Rotkreuz, Switzerland) with software version Version 1.5.0.39. PCR amplification was carried out in 20 µL volume reactions using the Lightcycler 480 Probes Master Kit (Roche Molecular Systems Inc., Pleasanton, CA, USA), containing 2 × Probes Master, 5 pmol of each primer, 2 pmol of the hydrolysis probe, and 5 µL of the template. For microbial pure cultures concentration of 2 ng μL^−1^ was used. The PCR program for the FusE qPCR consisted of an initial denaturation step for 10 min at 95 °C, followed by 39 cycles of denaturation for 10 s at 95 °C, annealing for 45 s at 57 °C, and extension for 1 s at 72 °C. In addition, negative controls with only PCR-grade water as a template were performed to rule out possible contamination. The *F. graminearum*-specific qPCR runs were carried out according to Yli-Mattila et al. [23], and *F. culmorum*-specific qPCR runs were carried out according to Waalwijk et al. [8].

The size of the amplified product was confirmed by loading the PCR products on 1.5% (*w*/*v*) agarose gel stained with Midori Green Nucleic Acid Stain (Nippon Genetics Europe, Düren, Germany) and using a DNA ladder (GeneRuler, Thermo Fisher Scientific, Waltham, Massachusetts, USA).

A standard curve was set up using a serial dilution of 0.001 to 1 ng target DNA per reaction of *Fusarium* DNA extracted from a pure culture of *F. culmorum* (VTT D-80148). The concentration of the *F. culmorum* DNA was quantified spectrophotometrically using a NanoDrop 2000 system (Thermo Scientific, Waltham, Massachusetts, USA). Three replicates were measured for each standard dilution.

The PCR inhibition of the cereal matrix was studied by spiking DNA extracted from a sterile oat sample with 1 ng of *Fusarium* genomic DNA. PCR reactions were performed in triplicate, as described above. The cycle threshold (Ct) values between the spiked reactions and *Fusarium* standard were compared to investigate potential PCR inhibition. The results obtained from the real-time PCR assay were analyzed with Abs Quant/2nd Derivate Max method in LightCycler^®^ 480 program (Roche Diagnostics Ltd., Risch-Rotkreuz, Switzerland), and the results were normalized by dividing the *Fusarium* DNA quantities with the amounts of total genomic DNA in the template.

### 4.6. Conventional Fusarium Plating from Field Samples

*Fusarium* contamination in cereal samples was determined by plating (100 grains/sample) on peptone-pentachloronitrobenzene (PCNB) medium (Nash and Snyder medium) [31] and incubating the plates in the dark at room temperature (22 °C). The growing colonies were isolated on potato dextrose agar (PDA), and *Fusarium* species were identified morphologically from the developing colonies. The percentage of grains contaminated with each identified *Fusarium* species was calculated.

### 4.7. DON Analysis

Trichothecenes were analyzed as described by Hietaniemi et al. [9]. The laboratories of Natural Resources Institute Finland apply a quality control system in accordance with SFS-EN ISO/IEC 17025:2017. In brief, 15 g ground cereal samples were extracted with 84% acetonitrile. The raw extract was purified with MycoSep #227 SPE column (RomerLabs, Getzersdorf, Austria). The cleaned-up extract was transferred to a silylated test tube and evaporated to dryness. Deoxynivalenol (DON) was identified and quantified as trimethylsilylether derivatives by GC-MS. The limit of quantification (LOQ) was 25 μg/kg. The method for trichothecenes (DON) has been accredited since 2003. The reference materials for DON quality control were either corn naturally contaminated with deoxynivalenol (Trilogy, TR-D100, 1.9 ± 0.1 mg/kg, Trilogy Washington, MO, USA) or matrix reference material deoxynivalenol in wheat (Biopure, BRM003022, 877 µg/kg ± 23 µg/kg and QCM2W1, 906 ± 68 µg/kg, Biopure, Cambridge, MA, USA).

### 4.8. Fusarium Community Analysis with NGS

DNA elutions extracted from 51 oat samples were sent to Illumina Miseq sequencing of the *Fusarium* elongation factor (*EF1α*) region to Microsynth AG, Switzerland. First, *Fusarium EF1α* libraries were sequenced with Nextera (Illumina, Inc., San Diego, CA, USA) two-step PCR with primers Fa-150 5′ CCGGTCACTTGATCTACCAG-3′ and Ra-2 5′ATGACGGTGACATAGTAGCG-3′ [19]. The libraries were sequenced at Microsynth AG by 2 × 250 bp paired-end sequencing on the MiSeq platform using MiSeq v3 Reagent Kit. Negative controls for amplicon barcoding PCR and positive controls including the known amount of DNA from species belonging to the genus *Fusarium* and a few other fungal species isolated from the cereal matrix were included in the sequencing set.

The sequence reads obtained from Illumina Miseq sequencing were subjected to sequence analysis using the DADA2 software package version 1.14 [32] and DADA2 Pipeline Tutorial 1.16 with some modifications. DADA2 package run in RStudio (RStudio, Boston, MA, USA, version 1.4.1106) with R version 4.0.4. First, the sequences were pre-filtered to remove ambiguous bases (Ns) that could affect accurate mapping. Then, primers were identified from the sequences and removed using the cutadapt tool [33]. The quality of the sequence reads was checked according to the DADA2 workflow. Next sequences were filtered and trimmed using DADA2 filterAndTrim function. Filtering parameters maxN = 0, maxEE = c(2, 2), truncQ = 2, minLen = 50 were used. A minimum length of 50 bp was used to remove spurious very low-length sequences. The maximum possible error rates were calculated using the learnErrors command. Identical reads were de-replicated (unique sequences). Amplicon sequence variants of the sequence data were identified using the DADA2 pipelines core sample inference algorithm. Denoised paired reads were merged according to the DADA2 pipeline, and an amplicon sequence variant table (ASV) table was constructed. Subsequently, chimeric sequence reads were removed from the dataset with the removeBimeraDeNovo function, using the consensus option. Finally, taxonomy from the domain to the species level was assigned to ASVs with DADA2’s native implementation of the naive Bayesian classifier method. Taxonomy was assigned against an in-house-generated *Fusarium* database for *EF1α* sequences. The *Fusarium* database for *EF1α* sequences was constructed according to Boutigny et al. [18], and in addition, sequences from Cobo-Díaz et al. [19] were added to the database. All images of the sequencing data were constructed with R using the packages phyloseq [34] and ggplot2 [35]. The Fusarium *EF1α* gene region sequences have been submitted to the European Nucleotide Archive (ENA, https://www.ebi.ac.uk/ena/, accessed on 31 July 2021) under accession numbers ERS7652901-ERS7652951.

### 4.9. Statistical Analysis

The correlation analysis to evaluate the correlation between qPCR results, DON content, and conventional *Fusarium* plating method was calculated using the IBM SPSS Statistics for Windows program (IBM Corp., Armonk, NY, USA, version 24). The Pearson or Spearman correlation coefficients at significance levels of *p* < 0.05 and *p* < 0.01 were calculated. Standard deviations of the qPCR results were calculated with Microsoft Excel (Microsoft Corp., Redmond, WA, USA, version 2102). Linear regression between Ct values and the amount of *Fusarium* DNA (ng) was calculated to acquire the r^2^ values of the qPCR standard curve. The qPCR, *Fusarium* plating method, and metabarcoding figures were created with R and correlation figures with Excel.

## Figures and Tables

**Figure 1 toxins-14-00045-f001:**
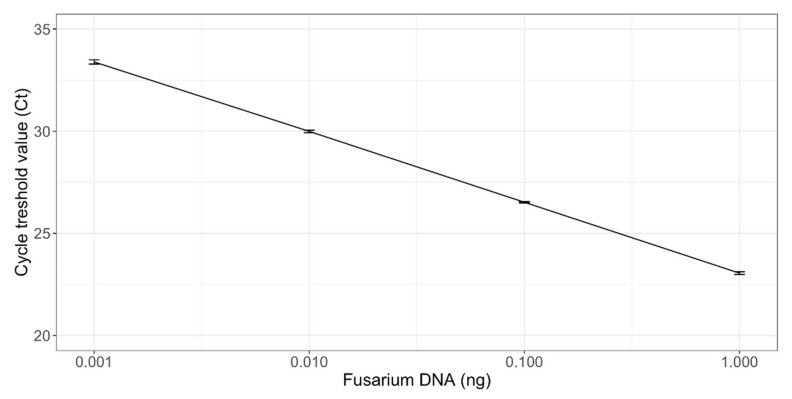
Standard curve of the FusE qPCR assay showing *Fusarium* DNA concentration (ng) against the cycle threshold (Ct) values of a single qPCR run. The *Fusarium* DNA concentration range was 0.001 to 1 ng per reaction. The amount of *Fusarium* DNA (0.001–1 ng) on the *X*-axis was plotted against the Ct values from 20 to 35 on the *Y*-axis. Linear regression equation of the standard curve was Y = −3.45x + 23.05 at r^2^ = 1. The efficiency was 95%. The standard deviation of three technical replicates is presented on vertical bars.

**Figure 2 toxins-14-00045-f002:**
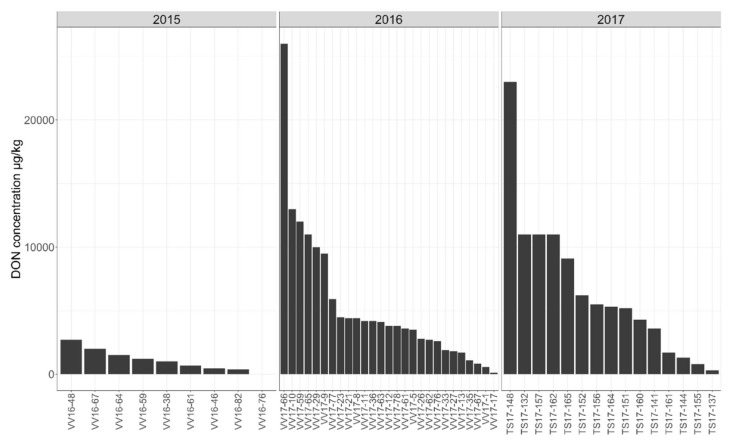
The content of mycotoxin deoxynivalenol (DON) in the oat samples in years 2015–2017.

**Figure 3 toxins-14-00045-f003:**
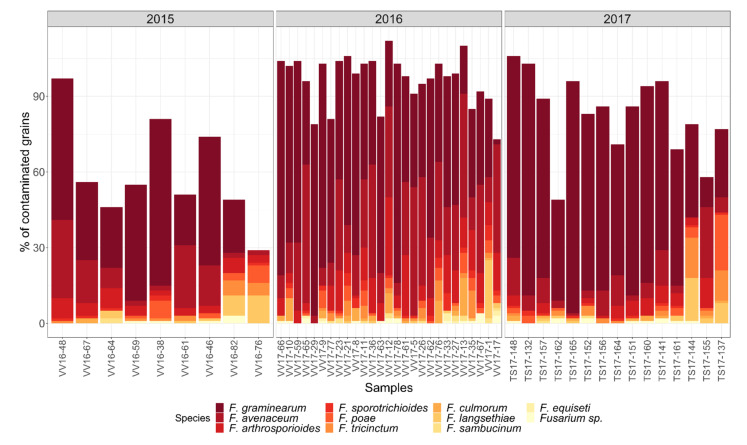
*Fusarium* species abundance in oat samples from the years 2015–2017 was detected with the plating method. *Fusarium* species abundance is shown as the percentage (%) of contaminated grains.

**Figure 4 toxins-14-00045-f004:**
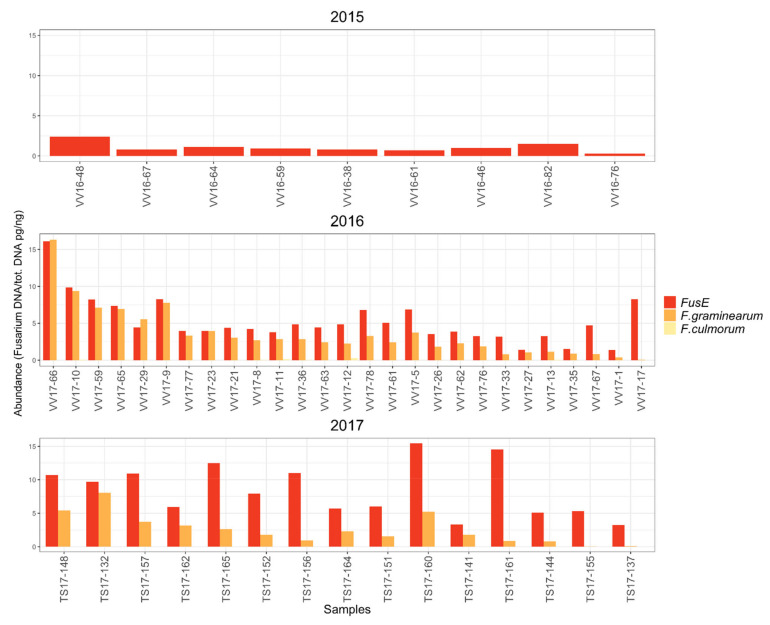
Amount of *Fusarium* spp. DNA, *F. graminearum* DNA, and *F. culmorum* DNA in the oat samples from the years 2015–2017. In 2015, only *Fusarium* spp. analysis was performed because contamination with *F. graminearum* and *F. culmorum* was low according to the *Fusarium* plating method.

**Figure 5 toxins-14-00045-f005:**
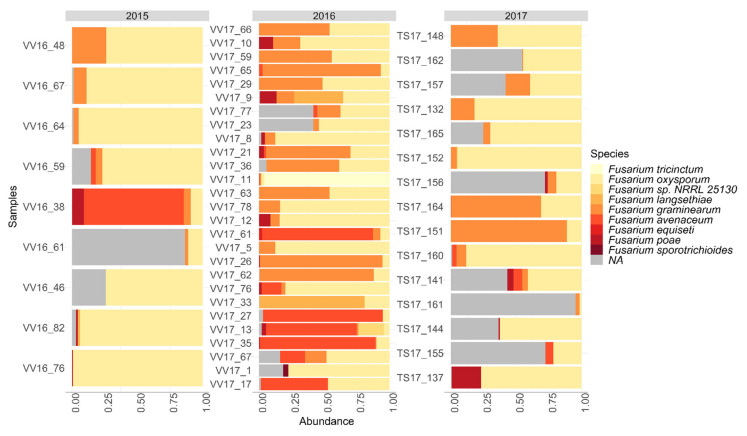
Relative abundance (%) of *Fusarium* species detected with the metabarcoding method in the oat samples in the years 2015–2017. NA: *Fusarium* spp., not identified to the species level.

**Figure 6 toxins-14-00045-f006:**
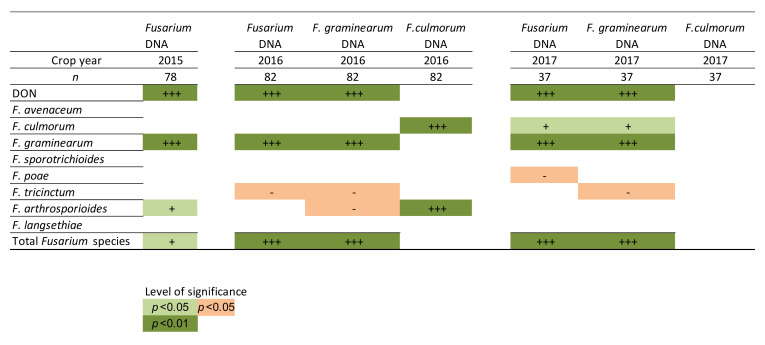
Correlation of the different *Fusarium* qPCR methods, DON content, and plating method of *Fusarium* species. *Fusarium* DNA indicates the results of the FusE qPCR assay and the *F. graminearum* DNA and *F. culmorum* DNA species-specific qPCR assays results. The correlation was analyzed by the calculation of Pearson or Spearman correlation coefficients at the significance levels of *p* < 0.05 and *p* < 0.01. The number of + and − signs indicates the levels of significance of the positive and negative correlation, respectively: + or −, *p* < 0.05, and +++, *p* < 0.01.

**Table 1 toxins-14-00045-t001:** Fungal and bacterial isolates and their origins.

Microbial Group	VTT-Strain	Species	Origin/Host, Country
***Fusarium*-species:**	VTT D-80141	*F. avenaceum*	Barley, Finland
	VTT D-96601	*F. cerealis*	Barley, Finland
	VTT D-80148	*F. culmorum*	Barley, Finland
	VTT D-82087	*F. equiseti*	Rotting fruit of *Cucumis melo*, Turkey
	VTT D-82082	*F. graminearum*	Barley, Finland
	VTT D-95470	*F. graminearum*	Corn, USA
	VTT D-03931	*F. langsethiae*	Barley, Finland
	VTT D-80134	*F. oxysporum*	Grain
	VTT D-76038	*F. poae*	Barley
	VTT D-82182	*F. poae*	Barley, Germany
	VTT D-77056	*F. sambucinum*	Grain
	VTT D-77057	*F. solani*	Grain
	VTT D-72014	*F. sporotrichioides*	Grain
	VTT D-131559	*F. tricinctum*	Barley, Finland
**Other fungi:**	VTT D-96653	*Achremonium polychronum*	Mouldy house, Finland
	VTT D-071272	*Aureobasidium pullulans*	Native wheat bran, Finland
	VTT D-76024	*Alternaria alternata*	Barley
	VTT D-00808	*Aspergillus ochraceus*	Barley, Finland
	VTT D-97673	*Clonostachys rosea f*. *catenulata*	Soil, Finland
	VTT D-76039	*Cochliobolus sativus*	Barley, Finland
	VTT D-03923	*Eurotium amstelodami*	Barley, Finland
	VTT D-94425	*Geotrichum candidum*	Malting process, Finland
	VTT D-131555	*Microdochium nivale*	Feed barley, Finland
	VTT D-131551	*Microdochium nivale*	Wheatfield soil, the Netherlands
	VTT D-131552	*Microdochium majus*	Winter wheat, USA
	VTT D-99750	*Penicillium verrucosum*	Barley, Denmark
**Yeasts:**	VTT C-92011	*Rhodotorula glutinis*	Barley malting, Finland
**Bacteria:**	VTT E-93497	*Leuconostoc citreum*	Malting process, Finland
	VTT E-90398	*Pantoea agglomerans*	Barley, Finland

**Table 2 toxins-14-00045-t002:** Sequences of the primers (FusEF, FusER) and probe (FusEP) of the FusE TaqMan qPCR to amplify the *Fusarium* spp.

Primer/Probe	Sequence (5′ to 3′)
FusEF (forward)	CTGGGTTCTTGACAAGCTCA
FusER (reverse)	CGGTGACATAGTAGCGAGGA
FusEP (probe)	TACCACGCTCACGCTCGGCT

## Data Availability

The NGS data that support the findings of this study are openly available in European Nucleotide Archive (ENA) at https://www.ebi.ac.uk/ under accession numbers ERS7652901-ERS7652951.

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
