# Peer review of "Taqman qPCR Quantification and Fusarium Community Analysis to Evaluate Toxigenic Fungi in Cereals"

_toxins, 2022, doi:10.3390/toxins14010045_

Round 1

Reviewer 1 Report

Evaluation and comments to the manuscript ID toxins-1524744, and entitled “Taqman qPCR quantification and Fusarium community analysis to evaluate toxigenic fungi in cereals”.

I share the authors' opinion that Fusariums are a serious problem in agricultural production, which contribute to quantitative and qualitative losses of crops.Moreover, some species are becoming more and more important in the infection of mammals, including humans.However, it should be remembered that some isolates, eg F. oxysporum, can be used in biological plant protection.Thus, this genus is very diverse and individual isolates within a species may play a different role in plant-fungal interactions as evidenced by the species F. oxysporum.

Overall, the manuscript is written in correct scientific language. I did not notice any major factual or editorial errors at work, and I do not mention any minor errors, because they are in some sense something normal in this type of work and they do not affect the general perception of work.

In my opinion, the experimental design and data analysis are appropriate and the introduction is correct. I think that we can always do something better, but this manuscript is on a good level. However, a few things need to be corrected in the manuscript before it is accepted for printing:

1) all Latin names must be italicized, including those in Figs.

2) please remove the dashes in the names of fungi in the Figs., and insert spaces between the individual members of the names of fungi, etc.

Reviewer 2 Report

Dear Authors,

Your manuscript is noteworthy, and the obtained results are of high scientific value both for development of TaqMan qPCR for Fusarium fungi on the genus level and the structure of Fusarium community in oat grain. The introduction is comprehensive, materials and methods are mainly well described, results are clear and fully discussed. The most of the comments relate to the style of presentation. I hope you find them useful.

In line 32 the lack of verb in a sentence: “from which most species belong to genus Fusarium and a couple of species belong to genus Microdochium

In line 36 and below: I believe the authors of fungal species should be cited.

In line 76 and line 371 the different gene abbreviation used (EF1α and EF1). Please, unify the abbreviation, and the gene name should be italic.

In line 101 there is a contradiction of the Microdochium species names (Microdochium nivale var nivale and M. nivale var majus) to the Table 1 (Microdochium nivale var nivale and M. majus) and discussion (just Microdochium nivale in line 241). Please, correct it. It should be Microdochium nivale and M. majus with species’ authors at the first mention.

In line 104, 122, 133, 372 the phrase “Error! Reference source not found” should be removed. Add references to figures that were meant.

In line 109, 116 and below Fusarium should be italic.

In line 120 and below replace “levels” and “concentration” with “content” for DON.

In lines 121-122 the thesis is understandable, but the sentence is incorrect, please rephrase.

In line 124 and 125 the double use of a word “significantly”.

I think the paragraph might be like this: “The deoxynivalenol (DON) content in oat grain varied substantially between the samples collected in different years. In 2015, the DON content ranged from 0 to 2700 µg/kg, and in average (N=9) was 1103 µg/kg. In 2016 and 2017, the higher DON contamination of grain was detected, and the content of this mycotoxin varied significantly between the samples. In 2016, the DON content ranged from 130 to 26000 µg/kg and in average (N=27) was 5334 µg/kg. Whereas in 2017, the DON content varied between 310 and 23000 µg/kg and in average (N=15) was 6620 127 µg/kg.”

In line 131 and below: I'm convinced that “by cultivation” is inappropriate method name. It should be “mycological method” or “mycological analysis”.

In line 136 the loss of preposition from before 2%.

In line 142 please rephrase “F. gramiearum abundance detected by cultivation”.

Line 155, 160 and below. It should be Fusarium spp. (not sp.) since it means the detection of all Fusarium fungi at the genus level.

In line 169 replace “taken” with “collected”.

In line 189 and 191 there are the mistakes in F. langsethiae species name.

In lines 191: “% but also F. graminearum was detected in the most of the samples (0- 88%) and was the dominant species in two of the oat samples. Other Fusarium species from the Fusarium genus detected in from the oat samples were F. poae, F. avenaceum, and F. langsethiae.”

Line 196, in diagram legend the strain NRRL 25130 is indicated as Fusarium sp. According Stepniewska et al., 2021 (https://doi.org/10.3390/f12060811) this strain was identified as Fisarium avenacium. Correct legend or comment this finding, please.

Line 212. I think it would be helpful to include the values of correlation coefficients to the figure.

In line 283 the phrase “only low amounts of F. culmorum were detected” is incorrect. Please rephrase.

Lines 304-307 represent the common place. This part should be removed.

In line 319 “levels” should be removed, “quantification of Fusarium DNA” is correct.

In line 335, 339 and below: the cultures of microorganism included in the study of FusE qPCR specificity were isolated, accurately identified and deposited to VTT collection. I think they should be mentioned as strains or pure cultures (as in lines 402 and 410), not isolates. Please, correct it.

In Table 1 Cucumis melo should be italic.

In lines 385-386 the two parts of table caption should be connected.

In line 402 “pure culture isolates” is incorrect. It should be “pure cultures” or “isolates”. The first is more consistent.

Line 425. Was the surface sterilization of the grains carried out before placing on the nutrient medium? Why was this nutrient medium chosen for the Fusarium fungi isolation?

In line 431: analysis

Best regards
